# Predicting Meridian in Chinese traditional medicine using machine learning approaches

Yinyin Wang[1], Mohieddin Jafari[1], Yun Tang[2], Jing Tang[1,3]*

**1** Research Program in Systems Oncology, Faculty of Medicine, University of Helsinki, Helsinki, Finland, **2** Shanghai Key Laboratory of New Drug Design, School of Pharmacy, East China University of Science and Technology, Shanghai, China, **3** Institute for Molecular Medicine Finland, Helsinki Institute of Life Science, University of Helsinki, Helsinki, Finland

* jing.tang@helsinki.fi

**Data Availability Statement:** All relevant data are within the manuscript and its Supporting Information files. The code is available at https://github.com/herb-medicne/meridian-prediction.

## Abstract

Plant-derived nature products, known as herb formulas, have been commonly used in Traditional Chinese Medicine (TCM) for disease prevention and treatment. The herbs have been traditionally classified into different categories according to the TCM Organ systems known as Meridians. Despite the increasing knowledge on the active components of the herbs, the rationale of Meridian classification remains poorly understood. In this study, we took a machine learning approach to explore the classification of Meridian. We determined the molecule features for 646 herbs and their active components including structure-based fingerprints and ADME properties (absorption, distribution, metabolism and excretion), and found that the Meridian can be predicted by machine learning approaches with a top accuracy of 0.83. We also identified the top compound features that were important for the Meridian prediction. To the best of our knowledge, this is the first time that molecular properties of the herb compounds are associated with the TCM Meridians. Taken together, the machine learning approach may provide novel insights for the understanding of molecular evidence of Meridians in TCM.

## Author summary

In East Asia, plant-derived natural products, known as herb formulas, have been commonly used as Traditional Chinese Medicine (TCM) for disease prevention and treatment. According to the theory of TCM, herbs can be classified as different Meridians according to the balance of Yin and Yang, which are commonly understood as metaphysical concepts. Therefore, the scientific rational of Meridian classification remains poorly understood. The aim of our study was to provide a computational means to understand the classification of Meridians. We showed that the Meridians of herbs can be predicted by the molecular and chemical features of the ingredient compounds, suggesting that the Meridians indeed are associated with the properties of the compounds. Our work provided a novel chemoinformatics approach which may lead to a more systematic strategy to identify the mechanisms of action and active compounds for TCM herbs.

**Funding:** This work was supported by the European Research Council Starting Grant agreement [grant number 716063]; the Academy of Finland Research Fellow funding [grant number 317680]; and Helsinki Institute of Life Science Research Fellow funding. Y.W was supported by the China Scholarship Council [grant number 201706740080] and the Finland EDUFI Fellowship [grant number TM-18-10928]. The funders had no role in study design, data collection and analysis, decision to publish, or preparation of the manuscript.

**Competing interests:** The authors have declared that no competing interests exist.

## Introduction

Single-agent drug discovery has often experienced low success rates which can be largely attributed to the lack of efficacy as well as unsatisfactory safety, especially when treating complex diseases such as cancer [1] and diabetes [2]. Recently, polypharmacology that involves multi-drug combinations acting on distinct targets has been proposed as a paradigm shift of drug discovery [3]. However, without a systems-level understanding of disease and drug interactions, it maintains a challenge to develop a valid strategy for the rational selection of drug combinations. In East Asia, plant-derived natural products, known as herb formulas, have been commonly used in Chinese Traditional Medicine (TCM) for disease prevention and treatment. Herb formulas often involve multiple bioactive components to produce synergistic effects in a personalized medicine manner, aiming for maximal therapeutic efficacy as well as minimal side effects [4]. For example, the Fufang Danshen Diwan (*Dantonic pill*), a botanical drug consisting of extracts of Danshen (*Radix Salviae Miltiorrhizae*) and Sanqi (*Radix Notoginseng*) is currently approved in 26 countries outside the USA for the treatment and prevention of chronic stable angina pectoris and other cardiovascular disease related conditions [5]. In this regard, understanding the bioactive components and their mechanisms of action for herb formulas might provide important insights on the rational design of multi-drug combinations for complex diseases [6, 7].

The prescription of herb formulas in TCM has been based on a holistic principle to model the human body as a miniature system that resemble the universe, which is composed of five interacting Elements (metal, wood, water, fire and earth) [8]. Similar to other schools of systems medicine, the cause of diseases or symptoms can be perceived as the loss of balance between these Five Elements [9, 10]. Treating a given disease is therefore equivalent to restoring the balance in the system [11], which can be achieved by either acupuncture [12, 13] or herb formulas that tune specifically certain inner channels of the body, known as Meridians [14]. There are 12 principal Meridians, each of which is linked to a specific TCM Organ and can be further attributed to one of the Five Elements (**Table 1**). The concept of Organ in TCM is fundamentally different from that of modern anatomic perspective, as the Organs in TCM represent certain distinct states of the human body, rather than a morphological structure. Similarly, although the Meridian system has been established as a fundamental basis of TCM several thousand years ago, it is not coincided to the known patterns of blood vessels or central nervous system [15]. More recently, fascia networks [16] and perivascular space [17] have been proposed to explain Meridian, but neither of them have been experimentally confirmed.

While the anatomical and physiological evidence of Meridians are yet to be determined, the narrative of TCM allows for the classification of herb formulas based on their targeting Meridians [18–20]. The rationale of Meridian has been investigated for a few TCM herbs. For example, Jie Geng (*Platycodi Radix*) has been considered as a Lung Meridian herb, and it was discovered recently that an active ingredient in Jie Geng called saponin can affect the lung and respiratory systems by the inhibition of lipid peroxidation [21]. Another example is Danshen, the dried root of *Salvia miltiorrhiza burge*, which has been used for treating cardiovascular diseases and hepatitis as a Heart and Liver Meridian herb [22]. Recent studies have shown that its lipophilic ingredients such as tanshinones and hydrophilic ingredients such as salvianic acids may play a synergistic role to achieve its therapeutic efficacy [23]. With the increasing knowledge about the biochemical and pharmacological properties of the bioactive ingredients from the TCM herbs, it is now possible to carry out a larger-scale analysis to investigate the molecular basis of Meridians and other concepts in TCM [24].

To leverage the complex biochemical and pharmacological datasets, systems biology approaches involving machine learning techniques have been utilized to the study of herb

**Table 1. The Meridians and their example herbs.** Each Meridian is linked to a particular Organ which is characterized by its Elements and Quality of Yin or Yang. TCM considers a disease a result of loss of balance in the Yin and Yang, which can be restored using herbs that target particular Meridians.

| Meridian name | Quality of Yin or Yang | Main Organ | Example herb |
|---|---|---|---|
| **Taiyin Lung Channel of Hand** | Greater Yin (taiyin) | Lung | *Rhizoma Pinelliae* |
| **Shaoyin Heart Channel of Hand** | Lesser Yin (shaoyin) | Heart | *Salvia miltiorrhiza* |
| **Jueyin Cardiovascular Channel of Hand** | Faint Yin (jueyin) | Cardiovascular | *Motherwort Herb* |
| **Hand's Minor Yang Three End** | Lesser Yang (shaoyang) | Three End | *Cape jasmine fruit* |
| **Taiyang Small Intestine Channel of Hand** | Greater Yang (taiyang) | Small Intestine | *Adsuki Bean* |
| **Yangming Large Intestine Channel of Hand** | Yang Bright (yangming) | Large Intestine | *Radix et rhizoma rhei* |
| **Taiyin Spleen Channel of Foot** | Greater Yin (taiyin) | Spleen | *Pueraria Root* |
| **Shaoyin Kidney Channel of Foot** | Lesser Yin (shaoyin) | Kidney | *Radix Angelicae Biseratae* |
| **Jueyin Liver Channel of Foot** | Faint Yin (jueyin) | Liver | *Bupleurum chinense DC* |
| **Shaoyang Gallbladder** | Lesser Yang (shaoyang) | Gall Bladder | *Spica Prunellae* |
| **Taiyang Bladder Channel of Foot** | Greater Yang (taiyang) | Urinary bladder | *Common Andrographis Herb* |
| **Yangming Stomach Channel of Foot** | Yang Bright (yangming) | Stomach | *Rhizoma Cyperi* |

formulas [25]. For example, Cheng *et al.* proposed a network-based methodology that can identify clinically efficacious drug combinations for specific diseases, which might be potentially used to explain also the pharmacology of TCM herbs [26]. Fang *et al.* summarized various chemo-informatics, bioinformatics and systems biology resources for reconstructing drug–target networks of natural products [27]. Fu *et al.* developed a data clustering method using a collection of 2,012 compounds associated with TCM herbs and discovered that the hot or cold nature of the herbs can be correlated with the physicochemical and target pathways of their ingredient compounds [28]. Wang *et al.* collected 5,464 compounds for 115 herbs and applied an unsupervised clustering method called Self-organizing map (SOM) to establish a classifier of cold and hot herbs based on the chemical structural fingerprints of the compounds [29]. However, these machine learning studies focused only on the hot/cold classification of TCM herbs, while it remains unknown whether the Meridian classification that involves 12 major classes can be also predicted from the chemical structure and physiochemical features of ingredient compounds.

In this study, we collected the Meridian information of herbs as well as the chemical structures of their ingredient compounds (**Fig 1**). These two datasets were utilized to determine the molecular features including structure-based fingerprints and ADME properties. With the feature matrices determined at both the herb level and the compound level, we further developed a machine learning framework to predict the Meridians of the herbs and their ingredient compounds. We tested multiple machine learning methods and showed that the classification of Meridians can be predicted especially at the compound level. These results suggested that Meridians indeed are associated with the molecular properties of herb compounds. We expected that our data integration approach may represent a novel perspective for the understanding of Meridian, which may ultimately lead to a more systematic exploration of the mechanisms of TCM.

## Results

### Distribution of Meridians at the herb level and the compound level

In total, 646 herbs including 10,053 ingredient components with Meridian and chemical structure information were obtained from the TCMID database (**S1 Table**). The Meridian distribution at the herb and the compound levels can be seen in **Fig 2**. At the herb level, altogether 333

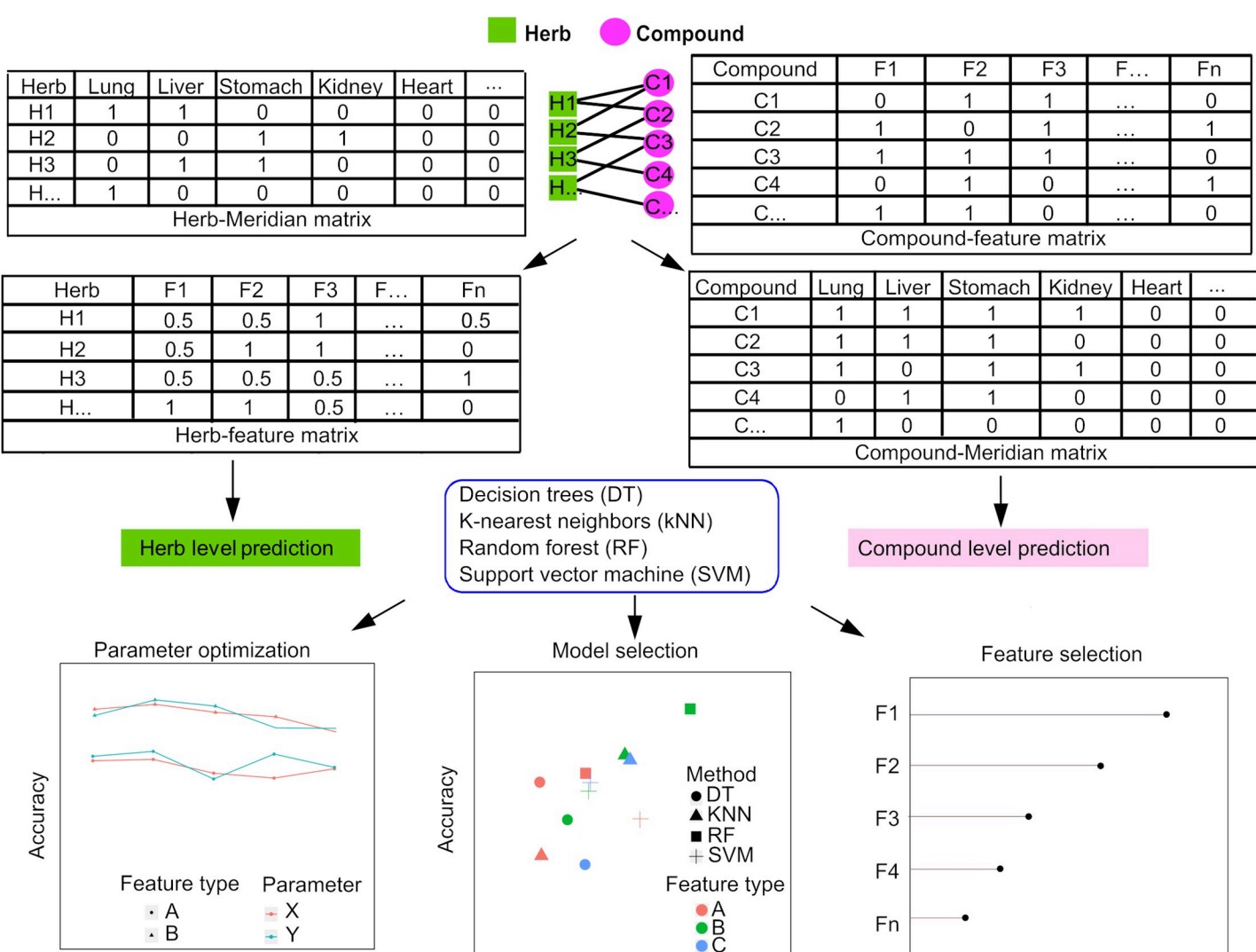

**Fig 1. Workflow of the study.** Herb-compound network shows the associations between herbs (green rectangles) and their active compounds (purple circles), which were used to determine the Herb-Feature and the Compound-Meridian matrices from the Herb-Meridian and Compound-Feature matrices. The features of herbs and compounds were determined from the chemical fingerprints and ADME properties. Machine learning methods were utilized to predict the Meridian classes for herbs and compounds respectively, by parameter optimization, model selection and feature selection.

herbs target the Liver Meridian, followed by Lung (n = 237), Stomach (n = 235), Spleen (n = 213), Kidney (n = 181), Heart (n = 155) and Large Intestine (n = 111) (**Fig 2A**). In contrast, much less herbs are found for the other five Meridians including Bladder (n = 57), Gallbladder (n = 33), Small Intestine (n = 24), Cardiovascular (n = 4) and Three End (n = 4). To avoid the over-interpretation of machine learning models on unbalanced datasets, we focused on the top seven abundant Meridians including Liver, Lung, Spleen, Stomach, Kidney, Heart and Large Intestine (**S2 Table**).

As expected, the majority of herbs (n = 580; 89.8%) target more than one Meridian, however, there is a varying degree of overlap between them. It can be seen that Kidney and Liver has the biggest number of shared herbs (n = 51), followed by 36 herbs that are common between Liver and Heart, and then 30 herbs between Liver and Stomach. The overlap between the Meridians illustrates the multi-target characteristics of TCM herbs. For example, Huo

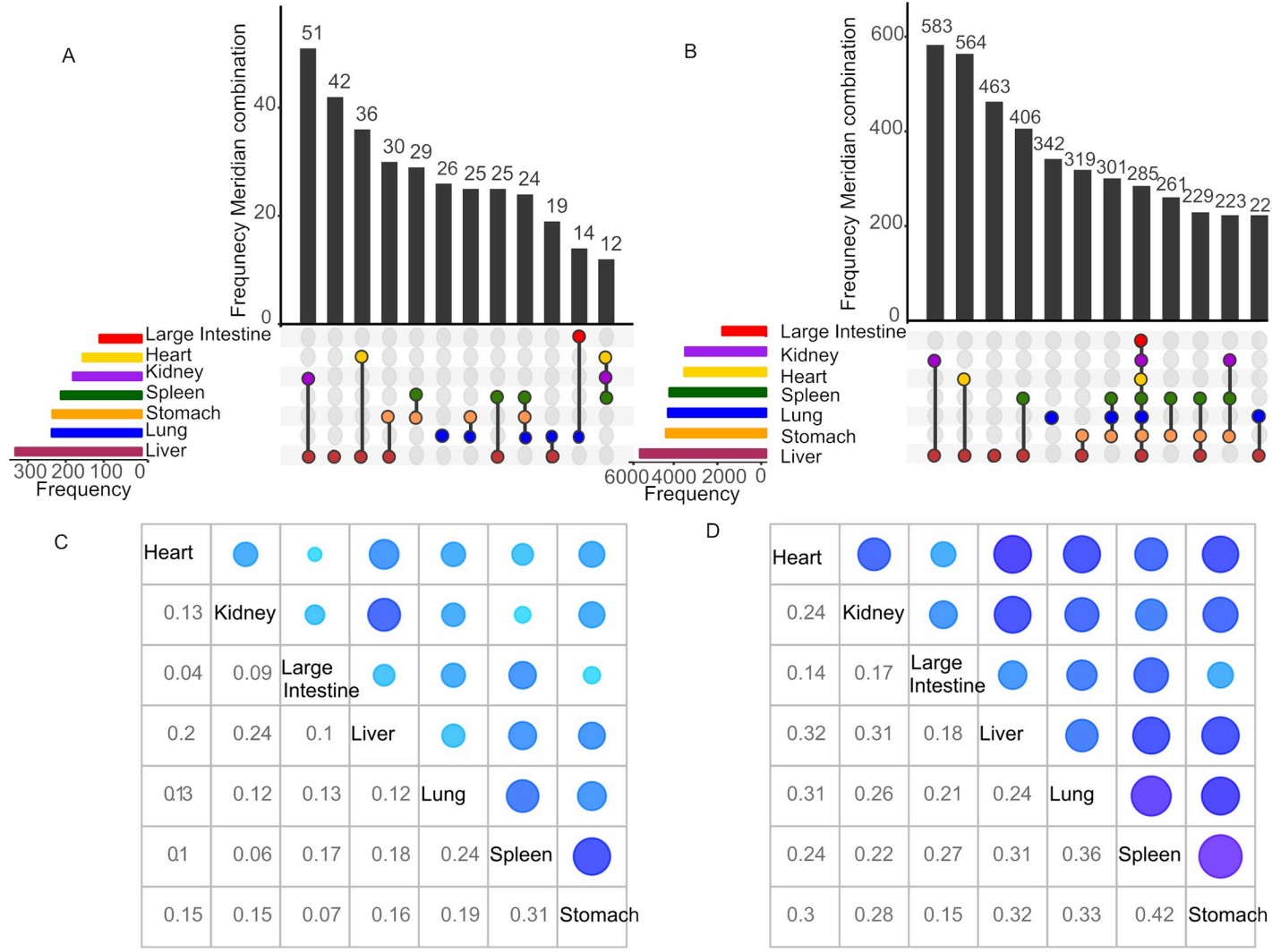

**Fig 2. Herb-Meridian and Compound-Meridian distributions.** (A-B) The color bars at the bottom left represent the frequency of herbs or compounds for each of the seven major Meridians, which can be further collapsed into subclasses depending on whether an herb or a compound is shared by one or several Meridians. The vertical bars show the frequency of herbs or compounds for a particular subclass of Meridian combination, as indicated by the connected lines below the x-axis between the Meridians. (C-D) The Jaccard coefficients between the Meridian pairs at the herb and the compound levels. The size of blue circles on the upper diagonal shows the degree of the similarity.

Xiang (*Agastache rugose*) belongs to Lung, Spleen and Stomach simultaneously [30], as this herb is known to relieve the symptoms of Lung, Spleen and Stomach diseases [31]. On the other hand, there are relatively fewer herbs that target only one Meridian. For example, 42 of the 384 (11%) Liver herbs are classified exclusively as Liver herbs and 26 of all the 260 (10%) Lung herbs do not target other Meridians. In contrast, all the herbs that belong to Stomach, Spleen and Large Intestine also target other Meridians. At the compound level, similar patterns was observed, where the Liver Stomach and Lung are again the top abundant Meridians (**Fig 2B**).

In order to quantify the overall similarity between these seven major Meridians, we calculated the Jaccard coefficients using the R package 'Corrplot' [32, 33]. The Jaccard coefficient, also known as Jaccard index, is a measure of overlap between two sets, with a value of zero for complete non-overlap while a value of one for identical sets [34, 35]. As shown in **Fig 2C and**

**2D**, the Jaccard coefficients between the Meridians are generally low, with the lowest score found between Heart and Large Intestine (0.04 at the herb level and 0.14 at the compound level), and the highest score found between Spleen and Stomach (0.31 at the herb level and 0.42 at the compound level). The average pairwise Jacaard coefficients are 0.15 and 0.26 for the herb level and for the compound level respectively, indicating that there are weak correlations between Meridians in term of the herb and compound distributions. Therefore, we considered the prediction of each Meridian separately in the following machine learning tasks. Ultimately, for a given new herb or a compound, its Meridians can be predicted using the best machine learning models.

### Prediction accuracy of Meridians using machine learning approaches

We carried out the prediction of the seven major Meridians at two data levels including herb level and compound level, for which their features were determined based on structure-based fingerprints and ADME properties. At the herb level, the ADME properties were also utilized to filter out those compounds with low water solubility or low gastrointestinal absorption (see Materials and Methods for more details). As a result, only 583 herbs remained after the filtering, covering 4,922 compounds. We evaluated the prediction performance under scenarios of different machine learning methods, feature types and data levels. More specifically, for each one of the seven Meridians, 84 machine learning-based models were constructed including all possible combinations from the four machine learning methods (SVM, DT, RF and kNN), seven feature configurations (Ext, PubChem, Sub, MACCS, ADME, Ext + ADME and All fingerprints + ADME) and three data levels (compound level, herb levels with or without ADME filtering). The model was trained by a five-fold cross validation using 70% data and then tested for its prediction accuracy using the remaining 30% data (see Materials and Methods for more details). To benchmark the model performance for each Meridian, we permutated the Meridian labels while keeping the ratio of positive and negative cases unchanged. The model performance for the permutated data was considered as the baseline.

As shown in **Fig 3A**, all the major Meridians achieved the top Balanced accuracy close to 0.65. Note that we pooled all the 84 machine learning models that differ in their feature combinations and machine learning methods, some of which were sub-optimal and therefore led to poorer prediction results. Still, these machine learning models performed significantly better than the baseline prediction of permutated models, in terms of Balanced accuracy and Matthews coefficient (**S1 Fig**, p-value < 0.0001, Wilcoxon rank-sum test). These results supported the general feasibility of using machine learning approaches to relate chemical information of herbs and compounds to explain Meridians (**S3 Table**).

Furthermore, using the Balanced accuracy metric, we found that the compound-level prediction performed significantly better than the herb-level predictions (**Fig 3B**, p-value < 0.001, Wilcoxon rank-sum test). The same trend has been observed by the AUROC and AUPRC metric (**S2 Fig** and **S3 Fig**, respectively). At the herb level, filtering out compounds with poor ADME properties improved the prediction significantly in Heart and Stomach (p-value < 0.05, Wilcoxon rank-sum test), while for Kidney, Lung and Spleen only the top machine learning models achieved higher prediction accuracy. In contrast, the ADME filtering seemed not helping the prediction of Large Intestine and Liver Meridians. In order to determine the chemical fingerprint features for an herb, we took the average of its compound features, based on the assumption that all the ingredient compounds are equally contributing to the pharmacology of the herb. This was likely an oversimplification of the actual mechanisms of action for a majority of herbs. However, the biological roles about the ingredient compounds were largely missing from TCMID and other resources, suggesting that the actual

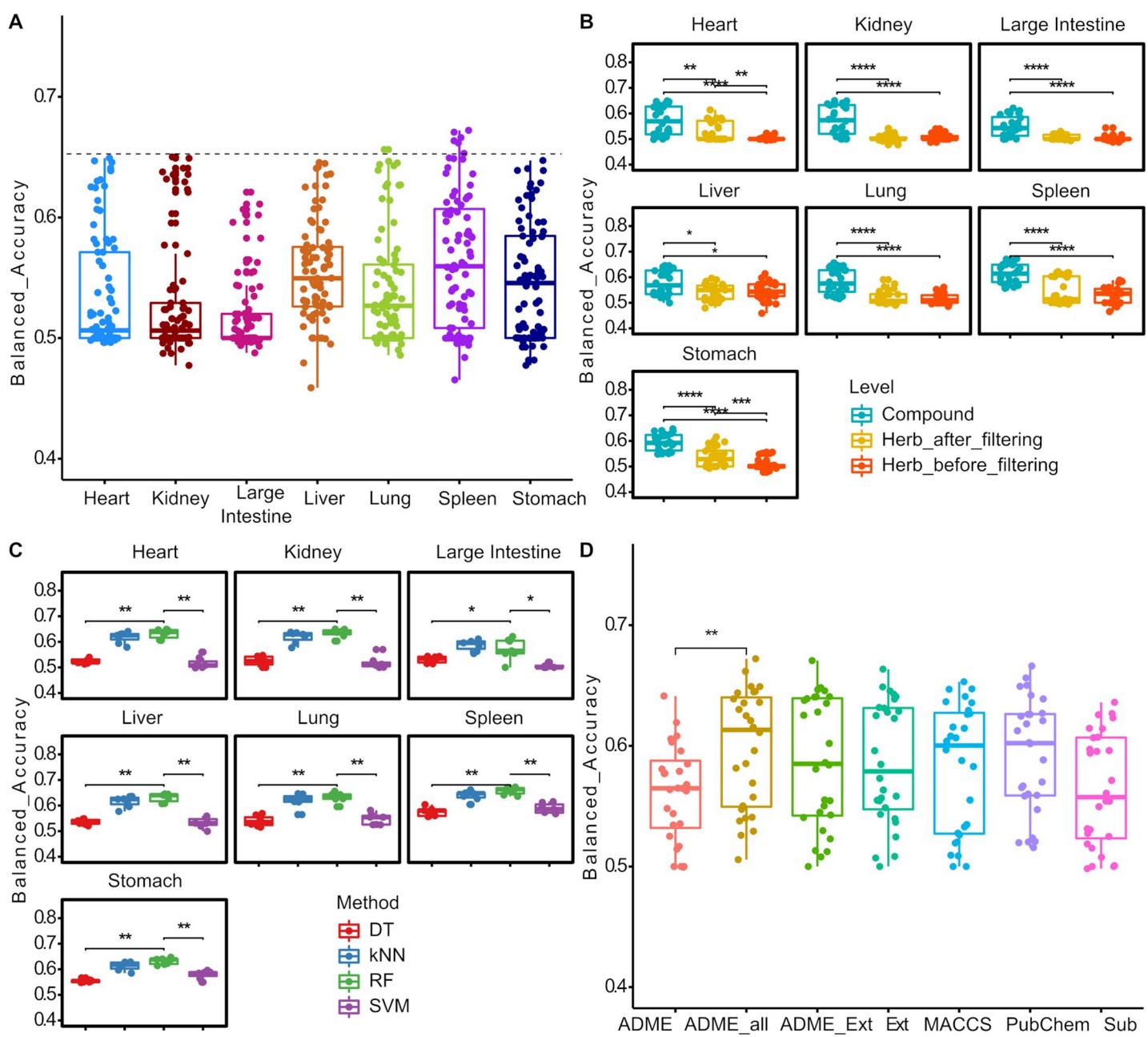

**Fig 3. Evaluation of the machine learning model predictions.** (A) The overall Balanced accuracy for the seven Meridians. Dashed line indicates the level of 0.65. (B) The Balanced accuracy at the three data levels (compound-level, herb-level before and after ADME filtering). (C) The balanced accuracy for the four machine learning methods at the compound level. (D) The balanced accuracy for the ADME and fingerprint feature types at the compound level. Wilcox rank sum test. *: $p < 0.05$; **: $p < 0.01$; ***: $p < 0.001$; ****: $p < 0.0001$.

contributions of these ingredient compounds have not been thoroughly resolved. In contrast, the compound-level data was more reliable, as each compound was treated independently when determining its molecular features and Meridians. This may explain the superior performance of compound-level predictions compared to the herb-level predictions. We anticipated that the herb-level prediction may be further improved when the actual composition and bioactivity of the compounds can be determined using modern high-throughput techniques e.g. mass spectrometry or HPLC (High performance liquid chromatography) [36].

As the compound-level prediction showed better performance than the herb-level prediction, we further compared the prediction accuracy between different machine learning methods at the compound level. As shown in **Fig 3C**, top models of RF performed better than kNN, DT and SVM across all the seven Meridians, suggesting that RF was able to detect the predictive features due to the use of ensemble learning techniques. We also evaluated the prediction accuracy of the machine learning methods using different feature types. As shown in **Fig 3D**, models with the different fingerprint types resulted in similar performance, while Ext and Pub-Chem fingerprints achieved the top Balanced Accuracy (0.67 and 0.66, respectively). This result was expected as the Ext fingerprint and PubChem fingerprint contains 1024 bits and 881 bits, respectively, which are the longer than MACCS (166 bits) and Sub (307 bits) fingerprint types. Furthermore, models using all the fingerprint types combined with ADME achieved higher top accuracies, compared to the use of ADME alone (**Fig 3D**). Taken together, we concluded that the combination of all fingerprints with ADME features may carry the most comprehensive information to predict the Meridians at the compound level, for which the RF method achieved the best balanced accuracy compared to other machine learning methods (**Table 2**).

## Important fingerprint and ADME features to explain Meridian at the compound level

After determining RF as the best model, we determined the feature importance score according to its contribution to the change of model prediction accuracy at the compound level: if the removal of a feature resulted in a much worse prediction by the model, then the feature will be given a higher importance score. We selected the top 30 most important features for each Meridian, resulting in 59 unique features in total, including 27 ADME properties and 32 fingerprints. We confirmed that the 59 important features were significantly more predictive than the other features across all the seven Meridians ($p < 0.0001$, Wilcoxon rank-sum test), with the median importance score for these 59 top features ranging from 2.77 for Large Intestine to 6.4 for Spleen (**Fig 4A**).

To evaluate the top features across the Meridians, we generated the bi-clustering heatmaps for the top ADME and fingerprint features separately. As shown in **Fig 4B**, lipophilicity features including iLOGP, WLOGP, MLOGP are among the top ADME features across all the seven Meridians, with the mean Z-score of feature importance of 1.66, 0.74 and 0.67, separately. This suggested that lipophilicity plays important roles for the Meridian classification of compounds. Molar refractivity (MR), a measure of the total polarizability of a substance, was identified as another important feature (mean Z-score 0.96). In addition, Solubility features predicted by the multiple methods using SwissADME have also shown relatively higher importance, with mean Z-scores ranging from 0.92 to 1.14. Lipophilicity is known to affect

**Table 2. The balanced accuracy that was achieved for each Meridian at the compound level by Random Forest using all the available features.**

| Meridian | Feature | Method | Balanced Accuracy |
|---|---|---|---|
| *Heart* | ADME + All fingerprint | RF | 0.65 |
| *Kidney* | ADME + All fingerprint | RF | 0.65 |
| *Large intestine* | ADME + All fingerprint | RF | 0.62 |
| *Liver* | ADME + All fingerprint | RF | 0.65 |
| *Lung* | ADME + All fingerprint | RF | 0.64 |
| *Spleen* | ADME + All fingerprint | RF | 0.67 |
| *Stomach* | ADME + All fingerprint | RF | 0.65 |

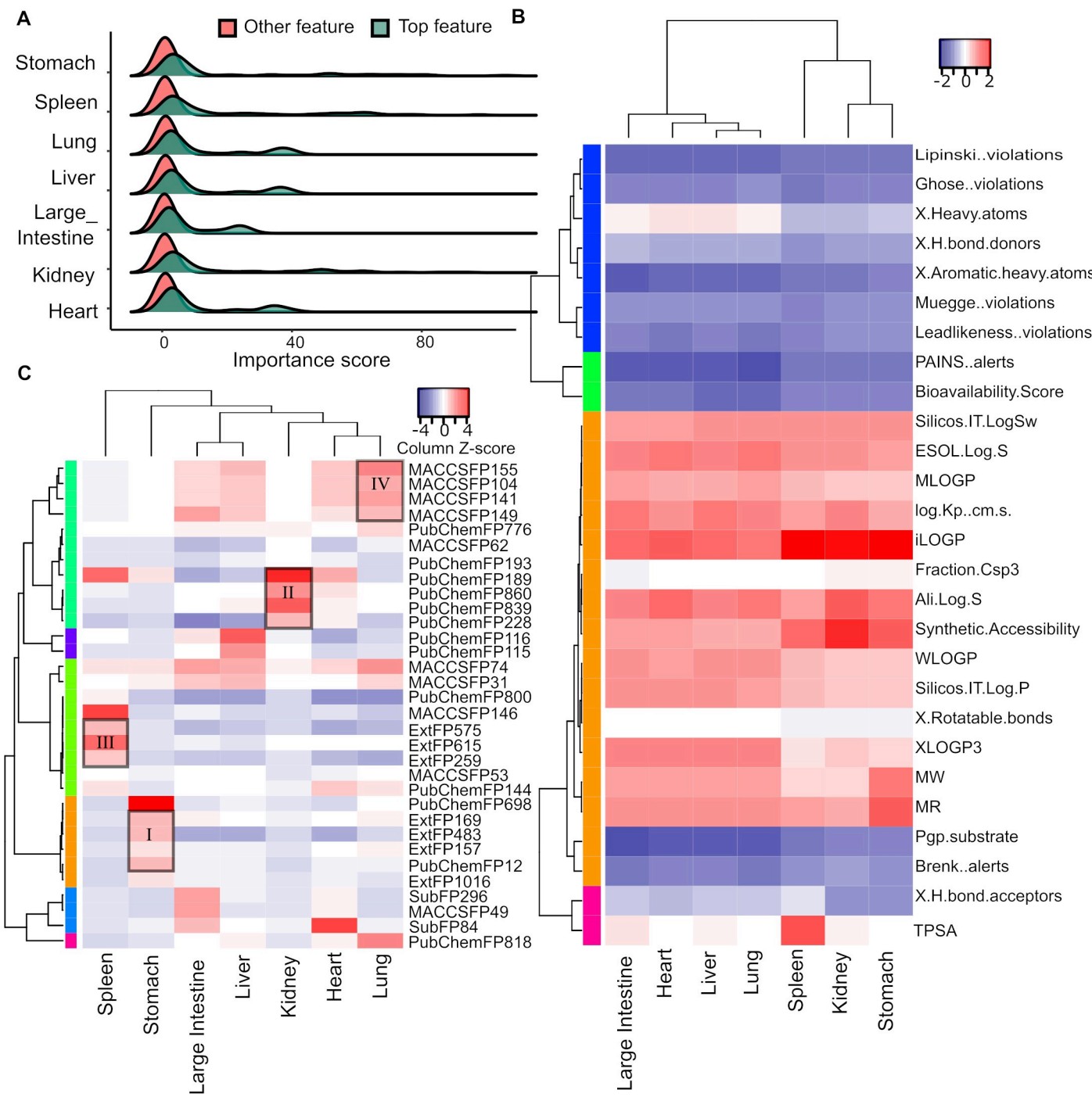

**Fig 4. Important features determined at the compound-level prediction of Meridian.** (A) The distribution of importance scores for the top 59 features as compared to all features. (B-C) The bi-clustering of the importance scores for the 27 ADME features and 32 fingerprints.

pharmacokinetic properties and the overall suitability of drug candidates [37]. Molar refractivity and Solubility are known to play important roles for the absorption and subsequent bioavailability of a drug in vivo. Our results suggest the rationale of including the ADME evaluation for understanding the pharmacology and pharmacokinetics of ingredient compounds in herb medicine.

We also evaluated the importance scores of the chemical fingerprints. As shown in **Fig 4C**, the fingerprint features from the same types tend to cluster together, with a Rand Index of 0.66 when comparing the similarity between the clustering by cutting the hierarchical tree at 1.5 and their actual feature types [38]. For example, the most important fingerprint features for Stomach Meridian formed a cluster (Cluster I in **Fig 4C**), which consisted of mainly Ext fingerprint features (Ext169, Ext483, Ext157 and Ext1016); The most important fingerprint features for Kidney are PubChem fingerprint features (PubChem228, PubChem189, PubChem839 and PubChem860) (Cluster II). Similar patterns were also found for Spleen (Cluster III as an Ext fingerprint dominant cluster) and for Lung (Cluster IV as a MACCS fingerprint dominant cluster). In general, the importance scores for the Ext fingerprints were higher among all the four fingerprint types (**S4 Fig**), which is also consistent with the better machine learning performances of Ext fingerprints described earlier in section 3.2 (**Fig 3D**).

Finally, we determined the important substructure fragments based on the top fingerprints. As shown in **S4 Table**, the representative fragments for each Meridian are quite different from each other, which is in line with the limited overlap of herbs between the Meridians (**Fig 2**). This result indicates that there might be enrichment of basic chemical structures that differs between Meridians, which can be further explored using pharmacophore modeling approaches [39].

## Discussion

Traditional Chinese Medicine (TCM) has gained increasing popularity in the drug discovery field, as shown by a few successful examples including the discovery of artemisinin for treating malaria and arsenic trioxide for treating acute promyelocytic leukemia [40]. Currently, there are around 1000 clinical trials on TCM herb medicine registered in the Clinicaltrials.gov [41] (retrieved in January, 2019), suggesting that the therapeutic potential of TCM has been actively researched through more rigorous scientific investigation. While the TCM theory is largely self-consistent as a philosophical narrative, the scientific rationale of why and how it is working remains elusive. For example, the interpretation of five elements and qi is rather metaphysical than physical, which makes many of the TCM concepts difficult to be translated into modern physiological and medical entities [9]. Furthermore, TCMs usually involve many active compounds that modulate various biological targets, where little is known about how these interactions lead to therapeutic relevance under a specific disease context. With the development of molecular profiling technologies, the extraction and characterization of the herb constituents is now possible and is expected to provide a comprehensive source of pharmacology data. Therefore, there have been strong needs for data integration to deconvolute the mechanisms of action of herb medicine in relation to the disease biology, so that a formal framework for testing and understanding of TCM can be established [42].

In this study, we built a computational framework to study the concept of Meridians, which has been long established for the classification of TCM herbs and thus constitutes the fundamental basis of treatment strategy in TCM. We collected the Meridian information for major TCM herbs and determined their features based on the chemical fingerprints and ADME properties. We found that an herb is commonly classified into multiple Meridians and that the correlations between them were generally low (**Fig 2**). Therefore, we decided to apply the one-vs-the-rest strategy to build classifiers for each meridian separately. Using supervised classification methods including Random Forests, Support Vector Machines, Decision Trees and K-Nearest Neighbor algorithms, we showed that the Meridians can be accurately predicted especially at the compound level, with a top balanced accuracy of 0.67 (**Fig 3**; **S3 Table**). Therefore, we concluded that molecular features of the compounds can be considered as the

essential information for an herb to be classified as a particular Meridian. In particular, we showed that the ADME properties improved the prediction accuracy, suggesting the relevance and reliability of the in-silico predicted ADME properties for the understanding of Meridians. For example, we found that Random Forests utilizing ADME features alone produced an AUPRC ratio of 2.29 for Large Intestine, topping the other Meridians, suggesting that indeed there is an evidence that ADME properties tend to be more predictive for this Large Intestine (Table 3). Ideally, experimentally-validated ADME properties for the ingredient compounds would be needed to confirm the prediction results. Furthermore, we considered 36 ADME features that were determined by SwissADME, assuming that TCM herb compounds become active when absorbed in the bloodstream. However, the therapeutic efficacy of herb medicine may be induced on gut microbiota, which do not necessarily interact with the bloodstream [43]. More relevant factors that may affect the ADME of herb medicine are expected to enhance the model prediction results. For example, another popular tool called admetSAR has been recently updated, which can provide 47 models for a more comprehensive evaluation of ADME [44, 45]. On the other hand, we evaluated four major structure-based fingerprint types, and found that their performances were similar. Despite that certain fingerprint types contain more bits than the others (e.g. 1024 bits for Ext fingerprint as compared to 166 bits for MACCS fingerprint), it seemed that all of them captured the essential structural information of TCM herbs and compounds.

We found that the compound-level prediction is in general more accurate than the herb-level prediction. There might be three reasons for that. Firstly, the exact compound composition for a given herb might not be accurate, as the extraction and detection of active components from herb medicine remains a challenge [46]. Secondly, even though certain compounds can be detected from a given herb, they may not be absorbable due to their poor ADME properties. As a result, the features that were determined for these compounds may play no therapeutic roles and thus do not affect the Meridian of the herbs. Thirdly, although the same compounds can be found from different herbs, their actual abundance may differ. In our construction of binary herb-feature matrix, there is lack of information to differentiate the different levels of compound abundance and their bioactivity. We expected the prediction accuracy at the herb level can be improved, providing that more accurate compound composition and activity data become available. In our modeling framework, the extraction of key features at the herb level can be done easily by first extracting the key features at the Compound level, and then combining them for a particular herb, using the Compound-Feature matrix and Herb-Compound matrix. With this framework, we may predict not only the Meridian for new herbs, but also for approved synthetic compounds for which their disease indications are already known. The link between Meridian and disease indications may provide more physiological understanding of Meridian.

**Table 3. The AUPRC ratio that was achieved for each Meridian at the compound level by Random Forest using ADME features only.**

| Meridian | Method | AUPRC ratio |
|---|---|---|
| Large intestine | RF | 2.29 |
| Heart | RF | 1.67 |
| Spleen | RF | 1.51 |
| Kidney | RF | 1.68 |
| Stomach | RF | 1.40 |
| Lung | RF | 1.45 |
| Liver | RF | 1.29 |

We identified that Random Forest (RF) as the best classification method, corroborating the superior performance of RF in similar machine learning tasks [47]. As an Ensemble Learning method, RF averaged the predictions from multiple decision trees and thus lowered the risk of overfitting. In the future, more advanced machine learning methods such as Deep Learning may be worth trying [48]. To make sense of TCM, the ultimate objective is not only a predictive model but also an interpretable model that can help understand the underlying mechanisms of action. Here, we identified the predictive features that may provide initial evidence for the molecular basis of Meridians, which may facilitate the discovery of novel active compounds from TCM herbs. As the main focus of our work is to provide the first evidence that machine learning approaches are feasible for interrogating the concepts of Meridians, we have not evaluated other more advanced methods including artificial neural networks. By further improving the knowledge of active ingredients for TCM herbs and the accuracy of machine learning algorithms, we expected that the machine learning framework can be greatly expanded towards a more systematic understanding of Meridians as well as other concepts in TCM.

TCMID is currently the largest database of TCM that collects over 49,000 prescriptions including 8,159 herbs and 25,210 ingredients. However, the majority of these herbs are lack of appropriate annotation on their Meridian information, highlighting the limited understanding of the topic. We extracted a subset of herbs from TCMID (n = 646) with known Meridian information and then included their ingredient compounds with known chemical structures (n = 10,053), with which the most predictive machine learning models and features were determined. To be able utilize our machine learning framework to predict the unknown Meridian for a given herb, the structural information of its ingredient compounds need to be provided as input data. With the structural information, it is then possible to determine the fingerprint and ADME features. In the future, we envisage that more comprehensive structural information about the active ingredients in herbs can be determined, so that the Meridian annotation of herbs can be done more systematically and more accurately. The advanced machine learning approaches that are tailored for analyzing such complex datasets may hold the key to the understanding of TCM rationale, which may ultimately provide novel insights for drug discovery and disease treatment [39].

## Materials and methods

The entire workflow of the present study was illustrated in **Fig 1**. First, herbs and their ingredient compounds were extracted from public databases. Molecular fingerprints and ADME properties were determined based on the chemical structures of the ingredient compounds, and were used to construct an Herb-feature matrix and a Compound-Meridian matrix. After obtaining all the features and Meridian classification for the herbs and the compounds, the prediction of Meridians at the herb and compound levels was implemented using four machine learning methods, including Support vector machine (SVM) [49], Decision tree (DT) [50], Random forests (RF) [51, 52] and K-nearest neighbor (kNN) [53]. The predication performance was further evaluated by cross-validation, based on which we identified the best models and feature types to predict the Meridians. The most predictive fingerprint features and ADME properties were identified for each Meridian separately.

### Data collection

**Meridian and ingredient compound information for TCM herbs.** We extracted the information of TCM herbs including the Meridian and the chemical components from the newly published database called TCMID [54], which is the largest database of TCM with over

49,000 prescriptions including 8,159 herbs and 25,210 ingredients. However, not all the herbs were included in our data analysis. As the aim of the study was to predict the Meridians based on the structural fingerprints of the herb ingredients, we focused on the herbs with known Meridian information from TCMID. Furthermore, for each herb we included only those ingredient compounds with known SMILES information, such that their structural fingerprints and ADME properties can be determined. The herbs with missing Meridian as well as missing chemical structure information of their ingredient compounds were discarded in this study. The curated dataset contained 18,140 herb-compound pairs including 646 herbs and 10,053 ingredient compounds.

**Chemical structural fingerprints for the ingredient compounds.** The canonical SMILES representations for the compound structures were determined using Open Babel [55]. We used the PaDEL-Descriptor software [56] to encode SMILES into a list of binary fingerprint features that indicate whether a particular substructure is present or absent in the compound. We considered four common fingerprint types including PubChem [57], MACCS (Molecular ACCess System) [58], Substructure (Sub) [59] and Extended fingerprint (Ext) [60]. PubChem fingerprint was extracted from the PubChem database (n = 881 bits) while MACCS fingerprint was originated from the cheminformatics system provided by the MDL company (n = 166 bits). Substructure fingerprint was used to represent the specific substructures based on SMARTS Patterns for Functional Group Classification (n = 307 bits) [59, 61]. Extended fingerprint complements the Substructure fingerprint with additional bits describing circular topological features (n = 1024).

**ADME properties for the ingredient compounds.** ADME properties play important roles to determine the pharmacokinetics of a compound, constituting the key factors that determine the hit and lead optimization processes in drug discovery. ADME properties describe how a compound deposits inside the human body in terms of the processes of absorption, distribution, metabolism and excretion. For instance, water solubility, usually measured as the decimal logarithm of solubility (log S) in the units of mol/l or mg/ml, indicates the maximum dissolvable concentration of a compound in water. After oral administration, a drug reaches the initial portion of the gastrointestinal tract, where the level of gastrointestinal absorption affects the fraction of the drug dose that enters the bloodstream. Lipophilicity, on the other hand, represents the affinity of a compound in a lipophilic environment and thus determines how easily the compound can pass through the lipid membrane of cells. For the TCM herbs, the ADME properties for their ingredient compounds have been largely uncharacterized. Therefore, we resorted to computational methods as an alternative, which have been shown previously to be able to reliably and efficiently determine ADME. For example, the Lipinski's Rule-of-five has been long used for evaluating the bioavailability based on the structure information of compounds [62]. Classical QSAR (Quantitative Structure-Activity Relationship) approaches also rely heavily on computational prediction of bioactivity properties based on the compound structures [56]. We determined the ADME properties of the ingredient compounds using an online tool SwissADME [63]. In the original publication, the authors of SwissADME showed that the prediction of Lipophilicity achieved an accuracy of r (correlation) = 0.72, MAE (Mean absolute error) = 0.89 and RMSE (root mean square error) = 1.14 against experimental data for 11,993 compounds. SwissADME also showed superior performance on the water solubility prediction with R2 (coefficient of determination) of 0.75, 0.69 and 0.81 based on three different models including the FILTER-IT model [63], the ESOL model [64] and the Ali model [65]. Notably, SwissADME has been recently applied to the study of plant-derived compounds including anticancer polyphenols from *Syzygium alternifolium* [66], PTPN1 (protein tyrosine phosphatase non-receptor type 1) inhibitors from several plant extracts [67] and a TCM called Zhi-zhu Wan [68]. Therefore, we considered the use of

SwissADME as a reliable method to probe the ADME properties for TCM herb compounds. The SMILES of each compound was loaded as input to SwissADME, and the result consisted of 36 ADME features including 6 drug likeness features, 5 lipophilicity features, 4 medicinal chemistry features, 9 pharmacokinetics features, 9 physicochemical properties and 3 water solubility properties (S5 Table).

## Construction of Compound-feature matrix and Herb-feature matrix

In this study, the features of a compound were considered as the combination of its fingerprint and ADME features, including 2378 fingerprint features (1024 Ext bits, 881 PubChem bits, 307 Sub bits and 166 MACCS bits) and 36 ADME property features. The four fingerprint types (Ext, PubChem, Sub and MACCS) were first evaluated separately in the machine learning models to determine the best fingerprint type. Then, we combined this best fingerprint type with the ADME features to check whether model performance can be further improved. The resulting Compound-feature matrix $X_C$ contained 10,053 rows of compounds and 2,414 columns of features.

Based on a previous study, a drug combination's molecular features can be represented by merging the features of its component drugs [69]. We considered also an herb as a mixture of different ingredient compounds, and determined the herb features as below:

Let $C_j = (c_1, c_2, ..., c_k)$ denote the set of ingredient compounds for herb $j$, where $k$ is the number of compounds. For each compound, its compound feature vector is denoted as $F_{compound} = (f_1, f_2, ..., f_n)$, where $n$ is the number of features. We modelled the herb feature $F_{herb} = (g_1, g_2, ..., g_n)$ as the average of its compound features, i.e.

$$g_{i, i=1, ..., n} = \frac{\sum_{c_1, c_2, ..., c_k} f_i}{k} \tag{1}$$

We collected 646 herbs and determined 2414 features including 2378 fingerprints and 36 ADME properties for their ingredient compounds. The Herb-feature matrix (HF) thus was size of 646x2414:

$$\mathbf{HF} = \begin{matrix} F_1 \\ F_2 \\ F_3 \\ F_4 \\ \cdots \\ {} \end{matrix} \begin{bmatrix} 0.2 & 0.1 & 0.3 & 0 & 0 \\ 0 & 0.1 & 0.1 & 0 & 0.8 \\ 0.1 & 0.6 & 0 & 0.1 & 1 \\ 0.5 & 0 & 0.1 & 0.3 & 0.1 \\ {} & {} & {} & {} & {} \\ 0 & 0.4 & 0.2 & 0 & 0 \end{bmatrix}_{646 \times 2414}$$

Furthermore, to evaluate whether filtering out the compounds with poor ADME properties affects the model prediction, we removed compounds that were predicted with logS lower than -6 by all the three water solubility models (the FILTER-IT model [63], the ESOL model [64] and the Ali model [65]) as well as low gastrointestinal absorption below 30%, which was a commonly accepted threshold to separate well-absorbed from poorly- absorbed compounds. After the filtering, 583 herbs and 4922 compounds were retained. We compared the model prediction accuracies before and after the ADME filtering.

## Construction of Herb-Meridian matrix and Compound-Meridian matrix

TCM herbs can be assigned to one or more of the 12 Meridians as shown in Table 1. For each herb, its Meridian vector is denoted as $M_{herb} = (m_1, m_2, ..., m_{12})$. From the 646 herbs that we

collected from TCMID, the Meridian classification for the herbs was represented as a binary Herb-Meridian matrix (HM) for the 12 Meridians as below:

$$\mathbf{HM} = \begin{array}{c} \\ M_1 \\ \\ M_2 \\ \\ M_3 \\ \\ M_4 \\ \\ \ldots \\ \\ \end{array} \begin{array}{cccccc} \text{Lung} & \text{Spleen} & \text{Stomach} & \text{Kidney} & \ldots \\ \left[\begin{array}{ccccc} 1 & 1 & 0 & 0 & 0 \\ 0 & 1 & 1 & 0 & 0 \\ 1 & 0 & 0 & 1 & 1 \\ 0 & 0 & 1 & 0 & 1 \\ 0 & 0 & 1 & 0 & 0 \end{array}\right]_{646 \times 12} \end{array}$$

We denoted that $\mathbf{H}_j = (h_1, h_2, \ldots, h_p)$ is a set of $p$ herbs that contain the compound $j$. The Meridian vector for this compound $\mathbf{M}_{compound} = (l_1, l_2, \ldots, l_{12})$ was determined as the union of the Meridians of the herbs in $H_j$, i.e.

$$l_{i,i=1,\ldots,12} = \mathrm{I}(\textstyle\sum_{h_1,h_2,\ldots,h_p} m_i > 0), \tag{2}$$

where $\mathrm{I}(\cdot)$ is an indicator function. The full Compound-Meridian (CM) matrix was constructed accordingly for the 10,053 compounds on the 12 Meridians:

$$\mathbf{CM} = \begin{array}{c} \\ C_1 \\ \\ C_2 \\ \\ C_3 \\ \\ C_4 \\ \\ \ldots \\ \\ \end{array} \begin{array}{cccccc} \text{Lung} & \text{Spleen} & \text{Stomach} & \text{Kidney} & \ldots \\ \left[\begin{array}{ccccc} 1 & 1 & 0 & 0 & 1 \\ 0 & 1 & 1 & 1 & 1 \\ 1 & 0 & 0 & 1 & 1 \\ 1 & 0 & 1 & 0 & 1 \\ 0 & 0 & 1 & 1 & 0 \end{array}\right]_{10053 \times 12} \end{array}$$

### Training the machine learning models

We set up the machine learning framework for each Meridian with binary response variables. Four supervised classification methods including SVM, DT, RF and kNN [70] were employed to predict the Meridians. These methods were implemented using the R package caret [71], with the default parameters listed in **S6 Table**. SVM is an algorithm which can determine a hyper plane to maximize the separation between the classes with minimal error. DT constructs a decision tree by representing an observation as a branch node and its classification result by a leave node. kNN is a distance-based learning algorithm where an object is classified according to a majority vote of its neighbors. RF is a decision tree-based ensemble learning approach where each tree votes for its preferred classification and the majority vote classification returns as the final prediction. We used five-fold cross validation to avoid overfitting when evaluating the model performance. Initially the data was split randomly to the training (70%) and testing (30%) sets. A five-fold cross-validation was applied to split the training data randomly into five equally sized folds. At each iteration, one unique fold was hold out while the remaining four folds were used to train a machine learning model. The model performance was then evaluated

on the hold-out fold. Such a process was repeated five times, after which the model that produced the highest accuracy was selected as the best model to predict the testing set, which comprise 30% of the total data. The model performance on the independent testing set was reported. The R scripts and input data for the machine learning framework are publically accessible at https://github.com/herb-medicne/meridian-prediction.

### Evaluating the prediction accuracy

We obtained a confusion matrix to evaluate the prediction accuracy for the test data. To avoid the inflated overall accuracy for imbalanced data, Balanced accuracy was also used to evaluate the performance of models, which is the average of sensitivity and specificity:

$$\text{Balanced accuracy} = \frac{\frac{TP}{TP+FN} + \frac{TN}{FP+TN}}{2} \tag{3}$$

True positive (TP) is the number of positive samples (*i.e.* herbs or compounds) which are correctly identified for a given Meridian; False positive (FP) is the number of positive samples which are not correctly identified. True negative (TN) is the number of negative samples which are correctly identified and false negative (FN) is the number of negative samples which are not correctly identified. Furthermore, Matthews correlation coefficient (MCC) and the Area Under the Receiver Operating Characteristic curve (AUROC) were also utilized for the model evaluation, defined separately as:

$$\text{MCC} = \frac{TP \times TN - FP \times FN}{\sqrt{(TP + FP)(TP + FN)(TN + FP)(TN + FN)}}, \tag{4}$$

and

$$\text{AUROC} = \int_{x=0}^{1} \text{TPR}(\text{FPR}^{-1}(x))dx. \tag{5}$$

The true positive rate (TPR) and false positive rate (FDR) were defined as $\text{TPR}(t) = \int_{t}^{\infty} f_1(x)dx$ and $\text{FPR}(t) = \int_{t}^{\infty} f_0(x)dx$ for a given classification threshold $t$, where $f_1(x)$ and $f_0(x)$ are the probability density functions for the predicted score for an instance if it belongs to positive and negative class, separately. Similarly, we evaluated the Area Under the Precision Recall curve (AUPRC) to focus on the prediction accuracy of positive cases.

### Identification of key features for the prediction of Meridians at the compound level

To find the most important features which play important roles for the Meridian classification, we used the varImp package [72] to estimate the variable importance based on the best models. Furthermore, the SARpy [73] tool was employed to detect key substructures (fragments) that emerge the most frequently as important features when predicting a specific Meridian. SARpy evaluates the significance of each substructure based on the likelihood ratio:

$$\text{likelihood ratio} = \frac{TP/FP}{P/N} \tag{6}$$

, where TP and FP stand for the number of compounds which contain the substructure and belong, or do not belong to the Meridian, respectively. We selected the top ten important substructures ranked by the likelihood ratio score for each Meridian. These substructures can be therefore considered as the most frequent fragments among the compounds of a specific Meridian.

## Supporting information

**S1 Fig.** Balanced Accuracy (A) and Matthews correlation (B) for all the machine learning methods on the real data as compared to permutated data at the compound and herb levels. ****: p-value < 0.0001.
(TIF)

**S2 Fig. Evaluation of the machine learning model predictions by AUROC (The area under the receiver operating characteristic curve).** (A) The overall AUROC for the seven Meridians. (B) The AUROC at the three data levels (compound-level, herb-level before and after ADME filtering). (C) The AUROC for the five machine learning methods at the compound level. (D) The AUROC for the ADME and fingerprint feature types at the compound level. Wilcox rank sum test. *: p < 0.05; **: p < 0.01; ***: p < 0.001; ****: p < 0.0001
(TIF)

**S3 Fig. Evaluation of the machine learning model predictions by AUPRC ratio, defined as the actual AUPRC divided by the baseline of random prediction.** (A) The overall AUPRC ratio for the seven Meridians. (B) The AUPRC ratio at the three data levels (compound-level, herb-level before and after ADME filtering). (C) The AUPRC ratio for the five machine learning methods at the compound level. (D) The AUPRC ratio for the ADME and fingerprint feature types at the compound level. Wilcox rank sum test. *: p < 0.05; **: p < 0.01; ***: p < 0.001; ****: p < 0.0001.
(TIF)

**S4 Fig. The importance scores grouped by the feature types according to Random Forest predictions for the seven Meridians at the compound level.**
(TIF)

**S1 Table. The Meridians and other TCM annotations for the 646 herbs.**
(XLSX)

**S2 Table. The numbers of positive and negative samples for each Meridian at the herb and the compound levels**
(XLSX)

**S3 Table. The prediction performances for the combinations of data levels, feature types and machine learning methods.**
(XLSX)

**S4 Table. Top 30 important ADME features, fingerprint bits and important substructure fragments for each Meridian determined at the compound level.**
(XLSX)

**S5 Table. The 36 ADME properties based on the chemical structure of compounds.**
(XLSX)

**S6 Table. Parameters of the machine learning models.**
(XLSX)

## Acknowledgments

We thank the authors of the TCMID database for making the herb medicine annotation data fully accessible.

## Author Contributions

**Conceptualization:** Yinyin Wang, Jing Tang.

**Data curation:** Yinyin Wang, Mohieddin Jafari.

**Formal analysis:** Yinyin Wang, Mohieddin Jafari.

**Funding acquisition:** Jing Tang.

**Investigation:** Jing Tang.

**Methodology:** Yinyin Wang, Jing Tang.

**Writing – original draft:** Yinyin Wang, Jing Tang.

**Writing – review & editing:** Yinyin Wang, Mohieddin Jafari, Yun Tang, Jing Tang.

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
