## [Decision Letter · Decision Letter 0]

12 Aug 2019

Dear Dr Tang,

Thank you very much for submitting your manuscript 'Predicting Meridian in Chinese Traditional Medicine Using Machine Learning Approaches' for review by PLOS Computational Biology. Your manuscript has been fully evaluated by the PLOS Computational Biology editorial team and in this case also by independent peer reviewers. The reviewers appreciated the attention to an important problem, but raised some substantial concerns about the manuscript as it currently stands. While your manuscript cannot be accepted in its present form, we are willing to consider a revised version in which the issues raised by the reviewers have been adequately addressed. We cannot, of course, promise publication at that time.

Sincerely,

Alexander MacKerell

Associate Editor

PLOS Computational Biology

Weixiong Zhang

Deputy Editor

PLOS Computational Biology

[LINK]

Reviewer's Responses to Questions

**Comments to the Authors:**

Reviewer #1: This research presents a very interesting idea: using traditional machine learning algorithms and fingerprints of chemical compounds in herbs to predict herbs' Meridians. The manuscript is well organized and clearly written, and a large amount of computation is performed.

However, I do see several major statistical issues.

First, since an herb could belong to multiple Meridians as clearly stated by authors, this is a typical multiclass classification problem. However, this is totally ignored by the authors. From the method descriptions, they seem to use one-vs-rest approach. The authors need to explain why one-vs-rest is a good approach for this multiclass classification problem.

Second, relevant to the first issue, neural network models could be used for this multiclass classification problem without such one-vs-rest approach. Have the authors tried any NN models? Did NN models perform poorly in comparison with these traditional algorithms?

Third, apparently, the data is quite unbalanced. Machine learning models are generally very sensitive to "unbalancedness" of the training data. Authors did not discuss this at all. Correspondingly, numbers of positive and negative samples should be added in "Supplementary Table 3.xlsx".

Fourth, for unbalanced data, AUC-ROC (the area under the receiver operating characteristic curve) is a widely used metrics; I highly recommend AUC-ROC numbers are calculated.

Reviewer #2: Wang and colleagues proposed a comprehensive machine learning-based study to predict Meridian in Chinese Traditional Medicine. They integrated multiple types of molecular fingerprints and ADME properties of active compounds in herbs. They then evaluated four different machine learning algorithms by combing different types fingerprints and ADME properties, which is quite a novel and comprehensive insight. Some machine learning models reveal good accuracy in predicting herb-Meridian associations in cross validation. This is an impressive study which offers powerful machine learning-based approaches for evaluations of Meridian by Chinese Traditional Medicine. The main findings are well presented and the manuscript is well written. Several specific comments may help improve the manuscript further.

1. The reviewers appreciated that the authors collected large-scale herbs with specific ingredients from database. However, each active ingredient has different concentration across herbs. The authors use equal weight (concentration) for each ingredient for calculation of molecular fingerprints and ADME properties to build machine learning models. This limitation has to be well explained or discussed in the revised manuscript.

2. The authors only evaluated accuracy only for machine learning models. Several comprehensive indexes, such as AUC (area under ROC), and precision-recall curves should be added.

3. The authors integrated both molecular fingerprints and ADME properties for building models. However, the reviewer cannot find how they integrated ADME properties in Figure 1.

4. The authors systematically evaluated four different machine learning algorithms in this study. More details of parameters of machine learning models are suggested to provided. For example, which k used for kNN, which function (kernel or linear) used for SVM, how many trees used for Random forest, etc. The authors may get better performance if they optimize tree number in random forest models.

5. The authors calculated ADME properties using public tools. One popular tool, admetSAR should be discussed.

6. It is impressive that the authors found that RF model shows the best performance for large intestine as several key ADME properties are highly correlated with large intestine. Could the authors evaluate the performance of RF models on large intestine using ADME properties only.

7. Several key refs related to polypharmacy (10.1038/s41467-019-09186-x) and polypharmacology of natural products (doi: 10.1093/bib/bbx045) should be discussed.

Overall, this is an interesting study, which offer powerful computational tools and models for systematic evaluation of Meridian-herb associations, an important, complex biomedical research question in traditional Medicine.

**Have all data underlying the figures and results presented in the manuscript been provided?**

Reviewer #1: Yes

Reviewer #2: Yes

PLOS authors have the option to publish the peer review history of their article (what does this mean?). If published, this will include your full peer review and any attached files.

Reviewer #1: Yes: yuhong wang

Reviewer #2: No

---

## [Decision Letter · Decision Letter 1]

20 Oct 2019

Dear Dr Tang,

We are pleased to inform you that your manuscript 'Predicting Meridian in Chinese Traditional Medicine Using Machine Learning Approaches' has been provisionally accepted for publication in PLOS Computational Biology.

In the meantime, please log into Editorial Manager at https://www.editorialmanager.com/pcompbiol/, click the "Update My Information" link at the top of the page, and update your user information to ensure an efficient production and billing process.

One of the goals of PLOS is to make science accessible to educators and the public. PLOS staff issue occasional press releases and make early versions of PLOS Computational Biology articles available to science writers and journalists. PLOS staff also collaborate with Communication and Public Information Offices and would be happy to work with the relevant people at your institution or funding agency. If your institution or funding agency is interested in promoting your findings, please ask them to coordinate their releases with PLOS (contact ploscompbiol@plos.org).

Thank you again for supporting Open Access publishing. We look forward to publishing your paper in PLOS Computational Biology.

Sincerely,

Alexander MacKerell

Associate Editor

PLOS Computational Biology

Weixiong Zhang

Deputy Editor

PLOS Computational Biology

Reviewer's Responses to Questions

**Comments to the Authors:**

Reviewer #1: The revision addressed my concerns and suggestions.

The accuracy and ROC numbers remain low using the common machine learning standards. However, the relationship between the meridians and compounds is expected to be very complex, and these numbers may still be meaningful. Similar problems will likely occur more frequently as people try machine learning models to more complex biological phenomena.

The thinkings behind traditional Chinese medicine are distinct from those in Western medicine, but in my opinion they complement each other well. Machine learning, in particular deep learning which could deal with more complex relationship, could be a powerful method studying complex biological systems such as herbal formulas.

I have two suggestions which may be helpful for authors' future researches. First, to further test the significance of the model, you could use bootstrap. Basically, randomly assign meridians for the used samples, perform the same procedure, calculate the same accuracy numbers, and then compute confidence intervals. Such confidence numbers could be more convincing supports for these models.

Second, as I said above, the relationship between meridians and compound structures is obviously very complex. From my experiences, deep learning models, if well constructed, could help even for the sample size of this study.

Congratulations for this interesting work.

Reviewer #2: The authors has addressed my concerns.

**Have all data underlying the figures and results presented in the manuscript been provided?**

Reviewer #1: Yes

Reviewer #2: Yes

PLOS authors have the option to publish the peer review history of their article (what does this mean?). If published, this will include your full peer review and any attached files.

Reviewer #1: No

Reviewer #2: No

---

## [Editor Report · Acceptance letter]

6 Nov 2019

PCOMPBIOL-D-19-01126R1 

Predicting Meridian in Chinese Traditional Medicine Using Machine Learning Approaches

Dear Dr Tang,

I am pleased to inform you that your manuscript has been formally accepted for publication in PLOS Computational Biology. Your manuscript is now with our production department and you will be notified of the publication date in due course.

With kind regards,

Matt Lyles
